# Assessing the Environmental Drivers of Lassa Fever in West Africa: A Systematic Review

**DOI:** 10.3390/v17040504

**Published:** 2025-03-31

**Authors:** Natalie A. Davis, Madeline A. Kenyon, Bruno M. Ghersi, Jessica L. Decker Sparks, Jonathon D. Gass

**Affiliations:** 1Center for Conservation Medicine, Cummings School of Veterinary Medicine, Tufts University, North Grafton, MA 01536, USA; natalie.davis@tufts.edu (N.A.D.); jessica.sparks@tufts.edu (J.L.D.S.); 2Department of Public Health and Community Medicine, School of Medicine, Tufts University, Boston, MA 02111, USA; madeline.kenyon@tufts.edu; 3Department of Infectious Diseases and Global Health, Cummings School of Veterinary Medicine, Tufts University, North Grafton, MA 01536, USA; bruno.ghersi_chavez@tufts.edu; 4Friedman School of Nutrition Science and Policy, Tufts University, Boston, MA 02111, USA

**Keywords:** Lassa fever, Lassa virus, biodiversity, conservation, land use, climate, ecology, transmission dynamics

## Abstract

The spread of Lassa virus in West Africa is reliant on the abundance and distribution of its rodent host reservoirs. While the impact of environmental change on viral spread has been studied for many zoonotic viruses, there is still a limited understanding of how seasonal impacts, land-use conversion, and biodiversity loss influence the expansion of Lassa virus among reservoirs. This systematic review synthesizes existing research on the association between environmental variables and Lassa virus circulation in West Africa to inform future research, public health interventions, and One Health policy. We searched international and African scientific databases using a set of pre-defined search terms to obtain publications reporting on Lassa virus in West Africa between 1969 and 2023. A total of 9465 articles were retrieved from this search and 70 studies met inclusion criteria for this review. Through systematic data extraction, we identified seasonal precipitation, land-use change, and host expansion as key environmental drivers of Lassa virus in reservoir hosts; however, we also highlight notable gaps in knowledge that limit our current understanding of these complex relationships. This review underscores the need for interdisciplinary research and strategies to mitigate the impacts of environmental change on Lassa virus transmission and protect vulnerable populations in West Africa.

## 1. Introduction

Lassa fever (LF), a viral hemorrhagic disease endemic to West Africa, is intricately linked to ecological, environmental, and climatic factors, underscoring the complex relationships between the disease, its reservoir hosts, and the environment. The etiologic agent for the disease, Lassa virus (LASV), belongs to the *Arenaviridae* family and is predominately transmitted to humans by the multimammate rat, *Mastomys natalensis*, through contact with urine, feces, saliva, and blood [1,2,3]. LASV infections in humans are largely dependent on the spatial distribution of *M. natalensis*, a known agricultural pest and commensal rodent species predominately found in human settlements, with a preference for scavenging in indoor environments and in surrounding agricultural cultivations due to the availability of food sources [4]. Understanding how environmental factors influence LASV host dynamics is critical for evaluating the disease’s epidemiology and developing effective public health interventions.

LASV was first discovered in the Borno State of Nigeria in 1969 and is now endemic to multiple West African countries, with notable past and current outbreaks occurring in Nigeria, Guinea, Liberia, and Sierra Leone [5]. Infected rodents and sporadic human cases have more recently been reported in Mali, Togo, and Benin, suggesting the endemic range may be expanding [6,7]. Critical disease underreporting has resulted in a distinct underestimation of infections and case fatality rates [8]. As a result, disease estimates rely on mathematical modeling studies, which have estimated that approximately 37.7 million individuals in 14 West African countries reside in areas with suitable conditions for zoonotic transmission of LASV, with a projected annual incidence of 897,700 human infections, half of which are predicted to occur in Nigeria [2,9]. Based on these projections, it is estimated that 18,000 people die each year from LF; however, global climate change, mounting evidence of new rodent hosts, urbanization and expansion of agricultural practices in West Africa threaten to expand the ecological niche for LASV and influence disease spread into previously non-endemic areas [2,10,11,12,13,14]. Lassa virus is categorized as a Category A bioterrorism agent by the US Center for Disease Control, and Lassa fever was listed as a World Health Organization priority disease for research in 2016 due to its status as a hemorrhagic virus capable of widespread death and disease [15,16]. LF was further determined to be a priority zoonotic disease for West Africa by the Economic Community of West African States (ECOWAS) [16]. Despite these critical calls for action, LF remains a crucially understudied hemorrhagic disease largely due to a lack of available diagnostic assays and limited clinical research infrastructure [6,17].

LF epidemics have historically been seasonally dependent, with human cases typically peaking in January–February during the dry season in West Africa, and secondary peaks occurring in March [11,18]. The seasonal epidemiology of LF is largely attributed to the population dynamics of the primary rodent reservoir, *M. natalensis*, with differing evidence on whether this is due to seasonal fluctuations of virus prevalence in the host species, increased rodent abundance indoors during the dry season, host breeding cycles, or a balance of all three factors [4,18].

Further, the transmission of LASV is influenced by land-use patterns, with *M. natalensis* populations predominating in rural human homes and surrounding agricultural fields and having a lesser abundance in forested or urban areas [8]. The distribution and LASV seropositivity of *M. natalensis* are strongly positively correlated with increased habitat fragmentation, agricultural expansion, and savanna conversion, indicating a high dependency on human settlements and vulnerability to habitat expansion due to increased human development [11,19,20]. Projected climatic and land-use changes in West Africa are expected to create more favorable habitats for *M. natalensis*, driving species growth and potentially increasing the risk of LASV spillovers to humans [19].

The mechanisms by which seasonal precipitation, land-use conversion, and host dynamics drive LF outbreaks in West Africa are not fully understood, highlighting a need for further research to explore these complex relationships. This study aimed to systematically synthesize and critically evaluate the environmental drivers of LASV in West Africa and identify knowledge gaps in current research to target future efforts. A thorough understanding of how ecological and environmental variables relate to seasonal outbreaks of LASV is essential for guiding future public health efforts in the face of climate change, landscape alteration, and human settlement expansion.

## 2. Materials and Methods

### 2.1. Literature Search and Eligibility Criteria

A systematic literature review of LASV in West Africa was performed by two independent reviewers in accordance with the Preferred Reporting Items for Systematic Reviews and Meta-Analyses (PRISMA) 2020 Guidelines (PRISMA 2020 Checklist available in Appendix A). Between January and February 2024, six databases were systematically searched: Google Scholar, PubMed, Web of Science, BIOSIS, Embase, and African Journals Online. The primary keywords “Lassa”, “virus” and “fever” were utilized in all search strings and matched with all combinations of secondary and tertiary keywords (Appendix A). Secondary search terms were chosen to extract relevant articles on previously identified environmental drivers of LASV emergence and included the following: “Environment”, “Biodiversity”, “Biodiversity loss”, “Land use”, “Land use change”, “Climate”, “Climate change”, “*Mastomys natalensis*”, “*Hylomyscus pamfi*”, “*Mastomys erythroleucus*”, “Ecology”, and “Host”. Tertiary search terms included known geographical occurrences of Lassa fever and West African countries, and included the following: “West Africa”, “Sierra Leone”, “Guinea”, “Liberia”, “Nigeria”, “Côte D’Ivoire”, “Central African Republic”, “Mali”, “Congo”, and “Senegal”. Given the vast number of search results generated by Google Scholar, only the first 400 search results were extracted for each search string. By extracting only the first 400 results, we were able to perform a robust but feasible literature search and ensure the most pertinent studies were included for review. This amount was chosen to ensure an equal sample was taken across all search strings, including larger searches that returned greater than 500 results, as well as smaller searches that yielded less than 500 returns. Article publication years were limited to 1969–2023 as LASV was not discovered until 1969 [5]. For quality control, a secondary PubMed search was performed by an independent researcher to ensure all articles meeting inclusion criteria were screened.

Grey literature, including non-peer reviewed articles, case reports, letters to the editor, editorials, and reviews that did not include secondary data analysis were excluded. Studies published outside of the included date range, studies reporting on imported cases of Lassa fever, and studies not pertaining to LASV in West Africa were excluded. To reflect the natural conditions of environmental disease drivers, experimental studies performed on laboratory subjects or rodents were excluded. Finally, articles without English translation were excluded. Peer-reviewed, original articles which reported on primary data were included. Articles that performed secondary data collection and analysis, including modeling studies, were included. Papers that performed meta-analyses were not included. We qualitatively assessed bias using the Modified Downs and Black Checklist to guide our evaluation of bias in included studies, a commonly used tool for assessing the quality of observational studies. As summary statistics or meta-analyses were not conducted, full bias scores of all included studies were not performed.

### 2.2. Selection

After duplicate studies were removed, the titles and abstracts of retrieved studies were examined by two reviewers for relevancy to the research aims. Relevant studies were those that reported on LASV with relation to natural host reservoirs, environmental transmission, biodiversity, or land use. Differing opinions regarding study screening were discussed and brought to consensus between the two reviewers (n = 227). If consensus could not be reached, articles were sent to a third independent reviewer (n = 5). Full-text articles were obtained for studies that met the initial screening criteria (n = 208) and were reviewed by the same two authors. Discrepancies were again resolved through discussion and brought to a third reviewer when necessary. In total, 70 studies were retrieved for data extraction (Appendix A).

### 2.3. Extraction and Data Synthesis

A standardized data extraction form was developed to systematically extract relevant data from the included studies (n = 70). The form was initially piloted by having the two reviewers extract data from the same five studies and then compare results to ensure cohesive data extraction. Studies were then divided and independently reviewed. The data extraction form retrieved study titles, first author, publication year, publication journal, inclusion criteria, and geographical information. Reviewers assigned a primary theme to each article to assist with data synthesis: virus host, seasonality, land use or biodiversity. The form was then further specified by theme to ensure all relevant data were extracted, including study aim, type of study, sampling methods, inclusion of serological and viral sampling, land-use change, and seasonal impacts.

## 3. Results

### 3.1. Search Results and Study Selection

A total of 9465 articles were identified by the search terms in electronic databases and 6427 duplicates were eliminated before screening (Figure 1). A pool of 2821 articles were eliminated after title and abstract screening, leaving 218 articles for full-text review. Of the 208 articles eligible for full-text review, 138 did not meet all the inclusion criteria, leaving 70 articles to be included in the review.

### 3.2. Characteristics of the Included Studies

Reviewed articles eligible for inclusion were published between 1969 and 2023. Of the included studies, 4.2% (n = 3) analyzed the entirety of continental Africa. Further, 1.4% (n = 1) analyzed sub-Saharan Africa and 7.1% (n = 5) analyzed all countries in West Africa and did not focus on a specific country. Within West Africa, 35.7% (n = 25) of studies reported findings from Nigeria, 35.7% (n = 25) from Guinea, 1.2% (n = 9) from Sierra Leone, 5.7% (n = 4) from Ghana and Mali, respectively, and 1.4% (n = 1) from Benin and Liberia, respectively (Figure 2).

The extraction form was utilized to separate included studies by environmental driver, including seasonality, land use, biodiversity and virus host species (Figure 3). The non-*M. natalensis* host category captures papers that analyzed LASV hosts, excluding the main reservoir species, *M. natalensis*. Studies that analyzed other potential rodent hosts, including *M. erythroleucus*, *L. sikapusi*, *H. pamfi*, *R. rattus*, and *M. musculus*, are represented in this category, as well as non-rodent hosts, such as mammals and avian species. Seasonality, land use, biodiversity impacts, and novel virus hosts were identified through the data extraction form as the most prominent environmental drivers of LASV spread in West Africa.

### 3.3. Seasonality

#### 3.3.1. Seasonal Impacts on LASV Prevalence in Host Reservoirs

Seasonal precipitation patterns were the leading environmental driver associated with LASV infections in *M. natalensis* [2,10]. Several studies investigated the impact of seasonality on LASV and IgG prevalence in animal reservoirs and humans, with varying findings. LASV prevalence in *M. natalensis* was found to be 2–3 times higher in the rainy season than in the dry season in Guinea [4]. Further, mathematical models of LF epidemics in Nigeria have predicted that LASV infections in rodents occur in the first half of the year, approximately January–June. The number of infected rodents is predicted to peak in May, the rainy season, and decrease in December, the dry season [21,22].

Of the included studies, two longitudinal analyses found no significant differences in IgG antibody prevalence in *M. natalensis* between the sampling seasons. For one of the two studies, this result may have been influenced by a small sample size (n = 35) and longitudinal sampling across both endemic and non-endemic areas, as well as across multiple years [23]. In this same study, when seasonality was analyzed with a larger sample size (n = 553) strictly within Guinea, LASV prevalence in *M. natalensis* was significantly higher during the rainy season [23]. Therefore, seasonal differences in IgG antibodies may vary by species and location and should not be averaged across localities and years.

In the second of the two studies, failure to obtain a statistically significant result in the second longitudinal study may also have been due to longitudinal sampling across multiple localities, species and habitats [24]. This finding suggests that the phenomenon of seasonal viral fluctuations is best analyzed with species-specific, cross-sectional methods. This is also supported by the geographical disparity in *M. natalensis* seroprevalence being as high as 52% in Mayo Ranewo, Nigeria, 17–26% for other Nigerian localities, and as low as 1% in Abagboro [23].

#### 3.3.2. Seasonal Impacts on Host Reproduction

*M. natalensis* breeding cycles have a strong seasonal trend and it has been hypothesized that fecundity may influence LASV infections in rodent populations, with reproduction being higher during the rainy season than in the dry season [25,26]. Only one included study investigated the relationship between fecundity and LASV prevalence in *Mastomys* and found no significant impact, leaving this to be a largely understudied topic [25].

After *M. natalensis* reproduce in the rainy season in Nigeria, there is a population boom in the dry season, introducing a large population of susceptible rodents to virus transmission [18]. When seasonal breeding is considered, mathematical modeling suggests that the number of infected rodents in Nigeria peaks in December, increasing the risk of LASV spillover to humans when rodents seek food inside homes during the dry season [18]. In contrast, another Nigerian study that also analyzed seasonal LASV trends predicted the number of infected rodents to be minimal in December; however, this model did not incorporate reproduction in their transmission model and predicted breeding to peak in the dry season, rather than the rainy season [21]. Incorporating *M. natalensis* reproduction into the LASV viral cycle has produced conflicting results. Further research would help to elucidate how the rodent breeding cycles influence viral spread, particularly in the context of the West African rainy and dry seasons.

Similarly to *M. natalensis*, *M. erythroleucus* was found to also primarily reproduce in the rainy season, creating a population increase and young age structure in the dry season [27]. However, *M. erythroleucus* have also been observed reproducing year round, while reproduction for *M. natalensis* is expected to almost completely halt from the late dry season (February) to the early rainy season (June) [26,27]. The relationship between secondary host breeding cycles, such as *M. erythroleucus*, and viral exposure has not been explored.

#### 3.3.3. Rainfall and LF Incidence

Seasonal precipitation patterns were the leading environmental driver associated with LASV outbreaks in humans among articles screened. While precipitation may not affect the transmissibility of the virus directly, seasonal rainfall patterns influence the migration of host reservoirs and are significantly correlated with annual incidences of LF in humans [10,21,28]. *M. natalensis* populations are highly constrained by climatic variables and prefer areas of high, variable annual precipitation, with peak rainfall occurring in August and not exceeding 1500 mm annually [2,11,19,29,30]. Areas in West Africa that have experienced LF outbreaks in humans have had higher annual precipitation than those without, with epidemic peaks in the west of Africa and declining in the more arid northeast [12,19,21,31].

When this phenomenon was analyzed at the ward level within Nigeria, the smallest administrative level in Nigeria, wards with confirmed LF incidents had significantly lower average precipitation than those without LF incidents [32]. This finding suggests that precipitation may not influence epidemiological disease trends when analyzed at the community level, and may require larger expanses of spatial data to determine trends.

There is a significant, negative correlation between the timing of LF emergence events and rainfall, aligning with the established trend of increased LF cases in the dry season, particularly in Nigeria [21,29,33,34,35,36]. Mathematic epidemiological modeling demonstrates that epidemics in Nigeria between 2016 and 2020 consistently peaked in February, the height of the dry season in the region [37,38]. However, smaller outbreaks also occur year-round in Nigeria, due to endemic circulation [11,37,39,40,41].

### 3.4. Land Use

#### 3.4.1. Impact of Households on LASV Transmission

Included studies determined that spatially autocorrelated clusters of seropositive *Mastomys* spp. are found more frequently in human houses in high-endemic villages in comparison to areas without LF endemics, indicating that household transmission of LASV constitutes a major disease pathway [42,43,44,45,46,47]. Additionally, LASV-infected rodents have also been shown to significantly cluster in refugee camps and student hostels, highlighting the risk that virus transmission extends beyond households to other types of anthropogenic settlements [48,49,50]. *M. natalensis* have a particularly dominant presence inside houses in West Africa, as evidenced by a study in Sierra Leone. Of 1490 small mammals captured, *M. natalensis* was the most commonly sampled species, with an overall prevalence of 23.8% (357 of 1490). Of the *M. natalensis* trapped inside homes, 28 of the 29 individuals tested positive for arenavirus IgG antibodies [24]. The transmission risk of LASV to humans was further confirmed in a 2021 study that isolated LASV RNA from *M. natalensis* feces, emphasizing the risk of transmission to humans within the household through rodent excrement [51].

Rodent infestations in homes in Guinea were also significantly correlated with high LASV antibodies in humans. In Gueckedou, study participants had a 14% antibody prevalence and 98.6% of respondents reported a rodent problem, whereas in Pita, Guinea, participants had a 2.6% antibody prevalence and 32.7% reported a rodent problem [52].

#### 3.4.2. Rodent Abundance Inside Houses

Rodent abundance inside human houses is significantly influenced by seasonal precipitation patterns, regardless of infection status. Spatial analyses reveal that LASV-positive rodents cluster inside homes during the dry season, likely driven indoors in search of food during the agricultural off-season [44]. Rodent trapping success was found to be highest inside houses at the start of the rainy season, as the previous dry season concluded. As the rainy season progressed, rodent trapping success significantly decreased from 40% in April to 14% in July and then 24% in October, signifying the rodent migration from inside home during the dry season to outside homes during the rainy season as crops become plentiful again [23,26,29].

Two of the included studies also investigated the impact of seasonality on *M. erythroleucus*, a less prominent host of LASV. These studies determined that this species was less commonly found inside houses in comparison with *M. natalensis*, and had the highest overall abundance and trapping success indoors during the late rainy season [26,27].

When compared to other trapping locations, such as cultivations, forests, and savannah, included studies reported overall trapping success of *Mastomys* spp. to be highest inside human houses, particularly for *M. natalensis* [1,4,53,54,55,56]. A small mammal trapping study conducted in Sierra Leone determined that *M. natalensis* had an overall species prevalence of 23.8%, with 92% trapped inside homes [24]. The species abundance predominated indoors even when compared to surrounding agricultural land or bush area [42,45,57]. Specifically, having holes in the home, the presence of rodent burrows, and being in a multi-room square building were significantly associated with having increased rodent abundance [58]. When compared to other rodent species in West Africa, movements into or out of houses were mainly observed for *M. natalensis* [57]. In contrast, *M. erthrythroleucus*, had a low presence in human houses and can be predominately found in the savannah, fallow lands and cultivations, as defined by areas of land that are actively used for farming or agriculture [26].

#### 3.4.3. Ecological Niche of LASV-Positive Rodents

Studies comparing the trapping success of seropositive *M. natalensis* outside of homes found that these rodents are most located in peridomestic settings, such as gardens and cultivated areas close to houses, and are less frequently found in cultivations and forests situated further from villages [1,13,19,43,53,59]. *M. natalensis* often move in between both locations and stay for extended periods of time indoors when food availability is limited outdoors [57]. For studies that did not include cultivations as trapping sites, or for areas where cultivations are not plentiful, the highest proportion of seropositive *Mastomys* spp. were found in savannahs, with forests being second, and coastal regions having the lowest percentage of seropositive rodents [19,53,57,60].

#### 3.4.4. Habitat Fragmentation

Habitat fragmentation, as characterized by perimeter area ratios, was determined to be positively associated with LASV exposure risk in humans [61]. Additionally, high levels of vegetation are shown to be associated with lower levels of LASV [32,61]. LF occurrence is positively associated with agriculture land-use as the host species have shown a preference for recently cleared land and pastures, raising concerns that as agricultural and anthropogenic expansion in West Africa continues, the ecological niche for LASV may expand beyond current endemic areas [10,11,12,13,19].

Rice storage facilities and rice harvest yields were also shown to significantly characterize locations of human LF outbreaks [19,62,63].

#### 3.4.5. Human LASV Differences in West Africa

The included studies determined that human LASV prevalence in endemic areas can vary across different landscapes. A study from Sierra Leone found notable disparities in human antibody prevalence, with 8% in a southern coastal village and 52% in an eastern province town characterized by tropical secondary forest and rich agricultural practices [45]. This trend was also determined in Guinea, where the difference in seropositivity was 11.9% in the coastal region and 59.6% in the forested region [61].

While Lassa fever is commonly regarded as a rural disease, there is conflicting evidence on this notion, suggesting a possible shift in epidemiologic transmission patterns. In Guinea, a difference in human LASV seroprevalence was observed between rural and urban areas, with higher prevalence in rural areas (12.9%) compared to urban (10.0%); however, it was also noted that sera from rural areas were less likely to be tested, revealing a possible sampling bias [64]. A small sample size (n = 11) from the 2016 outbreak in Nigeria revealed that eight (73%) of the confirmed cases resided in urban areas, while three (27%) were from rural areas [39]. This trend was further explored in a 2023 study that included data from the 2017–2021 outbreaks in Nigeria and 1057 laboratory confirmed cases. The average number of LF incidences in rural wards was significantly lower than in urban wards, where higher population densities may facilitate human-to-human transmission and improved access to healthcare [32].

### 3.5. Biodiversity Impacts on Lassa Virus

Two studies discussed the protective impacts of rodent species richness and biodiversity on LASV in West Africa [31,65]. Rodent species diversity was reported to be low in the LASV endemic areas of southern Mali, while species diversity was higher in northern Mali where the risk of LASV exposure is much lower [65]. While this study did not further analyze these findings, a 2021 modeling study concluded that rodent species richness had a significant negative association with LF emergence events, with areas of LF emergence having lower species richness than those without [31].

### 3.6. Other Hosts of Lassa Virus

#### 3.6.1. Rodent Hosts

*Mastomys* spp. rodents, particularly *M. natalensis*, were the most common species found in the captured rodent populations across many studies [24,42,55,57,66,67,68,69,70]. LASV-positive *M. natalensis* adults had significantly smaller bodies, smaller brain volume, and smaller reproductive organs [71,72].

Of the included studies, five identified *M. erythroleucus* as a host species for LASV, with RNA LASV samples confirmed in Sierra Leone and Nigeria, and IgG antibodies detected in Guinea and Mali [14,23,43,50,73]. Phylogenetic analyses of the LASV sequences identified in *M. erythroleucus* in Nigeria revealed that these sequences emerged much more recently compared to those detected in *M. natalensis*, with emergence dates of 2005 and 1961, respectively [50,74]. It was also found that all *M. erythroleucus* derived variants always had an LASV sequence from *M. natalensis* as their closest relative, concluding that virus exchange between these species is possible and poses a threat of expansion into non-endemic areas through species exchange [50,75].

In addition to the identification of *M. erythroleucus* as a new host, PCR testing also identified RNA LASV sequences in *L. sikapusi*, *H. pamfi*, *R. rattus*, *M. baoulei*, *Mus mattheyi* and *M. musculus*, and IgG antibodies against LASV were detected in *L. striatus*, *P. daltoni*, *M. minutoides* and *P. rostratus*, *P. misonnei* [7,14,23,42,43,54,56,73,76,77,78]. Evidence of LASV RNA has also been identified in the *Crocidura* genus, also commonly referred to as white-toothed shrews, and the *Tatera* genus, also known as gerbils. However, these have not been classified to the species level and only a very small number of individuals have been detected with the virus [23,42].

Outbreaks in Ghana and Benin, where *M. natalensis* is not the predominant species, suggested that another rodent host may play a role in transmission. *M. baoulei* has been hypothesized as the fourth host species due to the presence of LASV-positive individuals in Benin, Togo, and Ghana. However, LASV has only been detected in a small number of individuals in two species of pygmy mice, including *Mus baoulei* and *Mus mattheyi* [7,54,78]. The role of pygmy mice as hosts of LASV should be further explored to elucidate the role that these species play in viral transfer to humans. To this end, included studies in this review identify *M. natalensis*, *M. erythroleucus*, and *H. pamfi* as the rodent species hosts for LASV.

While this review did not systematically search for rodent control measures, included studies concluded that the use of continuous control or rodent vaccination are more likely to significantly reduce LASV spillover [79,80].

#### 3.6.2. Non-Rodent Hosts

Two included studies tested non-rodent animals to identify potential new hosts [81,82]. RNA LASV sequences were detected by PCR in multiple non-rodent, non-primate species; however, their involvement in LASV maintenance and transmission was not determined, requiring further research to analyze the infectivity and transmissibility amongst non-rodent hosts [81]. When primates were investigated, LF-specific antibodies were detected in a small number of Mona monkeys in Nigeria, suggesting that non-human primates are exposed to LASV, but the included study did not determine their role in virus transmission [82].

## 4. Discussion

### 4.1. Lassa Fever in a Changing Climate

#### 4.1.1. Seasonality

Based on the included studies in this review, we identified seasonal precipitation as the primary driver of LASV in West Africa. Human LASV outbreaks typically occur during the dry seasons in West Africa, as particularly evidenced by the 2016–2020 epidemics in Nigeria, and are significantly dependent on the spatial occurrence of the host species [11,18,19,22]. Further, temperature and precipitation are significant contributing factors influencing ecological suitability for LASV, with precipitation being the main contributing factor to LASV in *M. natalensis* populations [10]. The cause of seasonal fluctuations of LASV prevalence in rodent populations and subsequent impact on human epidemics is still poorly understood and renders LASV transmission reliant on the environmental suitability for the host reservoirs. Climate models forecast an expansion of suitable habitat for *M. natalensis* in West Africa due to rising temperatures and increased precipitation, underscoring concerns that the global climate change may amplify disease transmission risks [19,83].

Global climate change is projected to impact the spread of the disease, with approximately twice as many spillover events predicted in West Africa by 2070 [19]. Historic climate data in West Africa have demonstrated a trend of increasing temperatures and precipitation from 1983 to 2010, with Ghana, Côte d’Ivoire, Guinea, and Senegal experiencing the most significant temperature changes, ranging from a 0.2 °C–0.5 °C increase per decade and increased precipitation of approximately 0.2–1.0 mm/day per decade across the entire region [83]. The present-day warming and increased variability of precipitation are projected to be exacerbated by future climate models, leading hypotheses that the endemic regions for LASV will expand. Ecological niche modeling predicts that by 2070 most of the region between Guinea and Nigeria will become suitable for LASV, as well as several non-endemic regions, including Cameroon, DRC, and areas in East Africa [10].

#### 4.1.2. Growth of Agriculture

The prevalence of LASV host reservoirs is also influenced by land-use patterns. Habitat destruction and rapid human population growth in West Africa have led to the displacement of rodent species and forced them into closer proximity with humans. Included studies determined *M. natalensis* to have a higher abundance in human settlements and agriculture close to homes and found *M. erythroleucus* to prefer fallow lands and savannahs further from villages [26]. Rangeland and pasture coverage are a substantial factor influencing ecological suitability for LASV and projected deforestation and growth of agriculture threaten to expand the ecological niche of LASV [10]. Rapid population growth in West Africa is driving land conversion, particularly in areas suitable for rainfed agriculture, such as Sudanian and northern Guinean sub-regions, which constitute 93% of the transitions from natural to human-dominated land-cover types [84]. Substantial land conversion has also been observed in Nigeria and Sierra Leone. Nigeria’s savannahs have been cultivated by large-scale commercial farming, while the woodlands of Sierra Leone have been slash-and-burned to promote fallow and farmlands. The intensification of agriculture can be expected to increase as the population in West Africa continues to grow [84].

Human population growth in Africa is further marked by a distinct decrease in woody plant coverage due to agricultural expansion, with populations in Sub-Saharan Africa increasing by an average of 40 persons per km^2^ in areas where woody cover decreased due to agricultural expansion [85]. Areas of forest fragmentation have been shown to be hotspots for Ebola virus disease in Africa; however, the role of habitat fragmentation in LASV expansion is a critically understudied topic. One included study determined that fragmentation and the proportion of forests appear to affect exposure risk, but the role in virus circulation remains undetermined [61].

### 4.2. LF Host Expansion

Included studies identify *M. erythroleucus* as a secondary, less abundant host species for LASV. Though *M. erythroleucus* had a lower prevalence in trapping studies as compared to *M. natalensis*, the two *Mastomys* species occupy distinct ecological niches, suggesting that LASV has the potential to expand into new territories beyond the current range of *M. natalensis* [86]. While *M. natalensis* is considered a commensal species and dominates human-centric environments, *M. erythroleucus* is a generalist species and has greater abundance in natural habitats and fields [14,87]. The detection of viral RNA in the two *Mastomys* spp. species and LASV antibodies and RNA-isolates in several other rodent species provides evidence that the virus is capable of horizontal transmission amongst neighboring rodents. Further, a 2023 risk assessment ranked LASV the number one wildlife virus for risk of spillover based on a risk ranking system evaluating 31 host, virus and environmental risk factors [88]. While a few included studies discuss the inclusion of *M. erythroleucus* as a host, the role of this species in LASV transmission and spread is not yet known. The implications of host expansion and viral spread into non-endemic areas require deeper study to accurately target public health interventions. The promotion and implementation of vaccines for rodents and mammals as a prevention strategy should also be further explored [89].

### 4.3. Call to Action

Our review identified critical gaps in knowledge regarding the environmental drivers of LASV in West Africa. First, there is varying evidence on the seasonal fluctuations of LASV prevalence in host and human populations. Further research is required to comprehensively evaluate how seasonality impacts host ecology, food availability and reproduction and the subsequent influence on human epidemics. Second, climatic changes and agricultural growth in West Africa threaten to expand the ecological niche of LASV. Future climate projects are required to target areas of potential expansion and inform public health policy to prevent future epidemics. Finally, there is ample evidence to suggest that other rodent species are exposed to LASV and may be capable of transmitting the virus; however, current studies do not explore their role in the LASV transmission cycle. Additional research should explore how rodent reservoirs in West Africa are harboring and transmitting the virus between species, as viral adaptation and host switching suggest that LASV may be able to expand past its current range via new hosts.

Through extensive data extraction and literature searching, this review reveals the most critical gap in knowledge as the impact of biodiversity loss on LASV expansion and ecological dynamics in West Africa. West Africa has become a hub for agricultural development and human expansion in recent years, leading to a rise in deforestation and biodiversity loss [90]. As a result, understanding the potential implications of these environmental changes on viral transmission is critical for safeguarding human health. Despite the growing body of research on biodiversity loss and disease emergence, only two included studies investigated its effects explicitly on LASV. This phenomenon has been explored for West Nile encephalitis, Hantavirus pulmonary syndrome and Lyme disease, where biodiversity loss has been shown to increase rates of transmission [91]. The “dilution effect”, a commonly studied theory in ecology, suggests that a higher biodiversity often leads to lower infection prevalence in host species. Reservoir hosts, such as *M. natalensis*, tend to be generalist species and thrive in areas of human disturbance, while habitat specialists, such as predators, die out from a lack of complex habitat [92,93].

The “dilution effect” and the impact of biodiversity loss on viral transmission has been investigated for many infectious diseases. Notably, the dilution of Puumala hantavirus (PUUV) was tested in bank voles, the host of PUUV [92]. The field vole (*Microtus agrestis)* and the common shrew (*Sorex anaeus)* are competitors of the bank vole (*Myodes glareolus)* [92]. When the system of competitors and predators was altered, increasing the density of the competitors decreased infection probability [92].

The competitor and predator model requires further research for LASV hosts. There is growing evidence that the invasive black rat, *Rattus rattus*, is outcompeting *M. natalensis*, suggesting a possible impact on zoonotic LASV spillover. Capture sites in Sierra Leone and Guinea that had high *R. rattus* captures had lower *M. natalensis* catches per trap [94]. Zoonotic disease spillover risk was also negatively related to the presence of *R. rattus*, while no other rodent species had a similar impact on catch per trap of LASV-positive *M. natalensis*. Further, the mean *R. rattus* presence effect was larger in magnitude when compared to seasonality effect within the fitted models, suggesting that *R. rattus* presence may be even more influential on *M. natalensis* abundance than seasonality. Further research is needed to elucidate the exact mechanisms by which *R. rattus* may influence *M. natalensis* populations and subsequently impact LASV spillover; however, these preliminary data exemplify the potential impact of biodiversity, as well as competitor ecology, on zoonotic disease spillover.

### 4.4. Study Limitations

This systematic review had several important limitations. First, due to journal accessibility and language restrictions, our literature search only included one database search of African origin. African Journals Online (AJOL) was chosen due to its global impact, commitment to the amplification of African research and diverse inclusion of many open-access, African journals. However, as a result, we may have missed papers not published in the journals included through AJOL.

Second, our search strategy only included articles with an English translation and may have missed relevant papers published in other languages. We chose to include the term “host” and not “reservoir” in our search strategy given that hosts include reservoirs as well as secondary hosts; however, it is possible that this decision affected the number of studies included in the systematic review. Third, we limited our Google Scholar extractions to the first 400 articles returned. This quantity was chosen to encapsulate search returns of all sizes; however, this strategy may have missed important articles that were not captured in this amount.

Fourth, many included studies focused on specific countries or regions in West Africa, particularly those with a higher socioeconomic standing and access to research facilities. This geographical bias may not be representative of the entire region. Further, our review focused on environmental drivers of LASV and may have missed other contextual factors that influence viral transmission, including socioeconomic conditions, human behavior, and healthcare systems. While these variables can have crucial impacts on viral expansion, it was beyond the scope of this review to include in the search strategies and analyses.

Finally, biases were considered in the interpretation of results from all included papers. Included papers were evaluated for inherent biases, including confounding, exposure/case selection, classification of interventions, missing data, measurement of outcomes, and selection of reported results. Papers with identified biases were discussed by the study team and our protocol was to remove inclusion eligibility for any paper that included biases that may have impacted reported results.

## 5. Conclusions

This systematic review provides a comprehensive overview of the environmental drivers of LASV in West Africa. Climate models predict an expansion of suitable environmental conditions for *M. natalensis* due to rising temperatures and increased precipitation, while rapid population growth in West Africa is influencing the conversion of natural habitats to agricultural lands and threatening to expand the ecological niche of LASV host reservoirs. The mechanisms by which seasonal precipitation, land-use change, and host dynamics drive human LASV epidemics in West Africa are not fully understood, highlighting a need for further research to guide future public health efforts and targeted research. Ultimately, this review underscores the urgent need for interdisciplinary research and proactive preventative strategies to mitigate the impacts of environmental change on LASV transmission and protect vulnerable populations in West Africa.

## Figures and Tables

**Figure 1 viruses-17-00504-f001:**
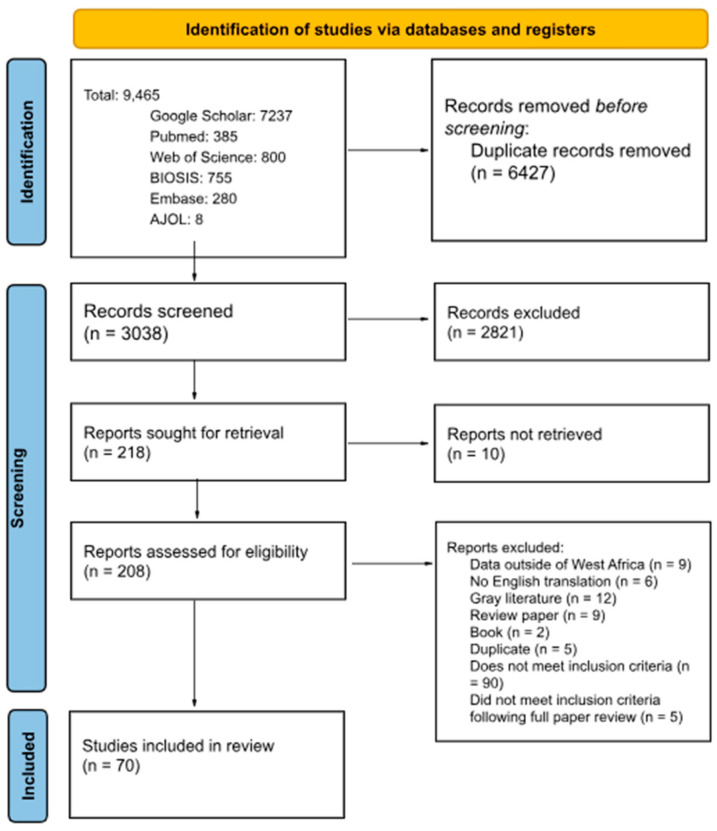
Preferred Reporting Items for Systematic Reviews and Meta-Analyses (PRISMA) flow chart of article selection. PRISMA Checklist can be found in Appendix A.

**Figure 2 viruses-17-00504-f002:**
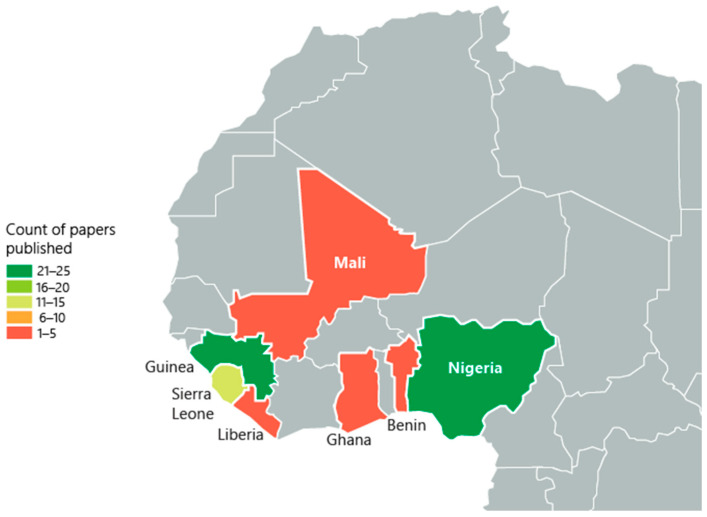
The geographic distribution of included studies with published data in West Africa. The numbers on the map represent the quantity of studies conducted in each country, highlighting regional research focus areas. Included countries: Nigeria (n = 25), Guinea (n = 25), Sierra Leone (n = 4), Ghana (n = 1), Mali (n = 1, Benin (n = 1), Liberia (n = 1)).

**Figure 3 viruses-17-00504-f003:**
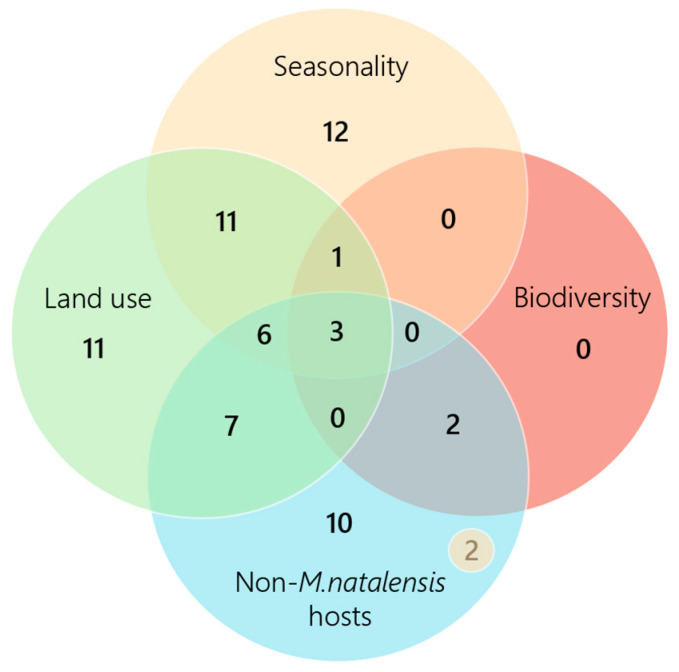
Number of papers that discuss each environmental driver. Biodiversity impacts were the least often analyzed, with six papers in total discussing this theme. Five included studies were omitted from the figure for not explicitly discussing one of the drivers. N = 2 included in the “Non-*M*. *natalensis* hosts” category represents the two papers that discussed this theme and seasonality.

## Data Availability

Data are available upon request to the corresponding author.

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
