# Peer review of "Assessing the Environmental Drivers of Lassa Fever in West Africa: A Systematic Review"

_viruses, 2025, doi:10.3390/v17040504_

Round 1
Reviewer 1 Report
Comments and Suggestions for Authors
The article reviews the evidence available on Lassa fever and environmental, ecological, and climate (change) associations with incidence prevalence in West Africa, from 1969-2023. The methodology is sound and clearly described. The results are easy to follow because these are presented each under its subheading and contrasting findings from the included articles are nicely described and discussed. The review provides a comprehensive summary of what information is available, contradicting and same findings, and the research gaps that need to be filled to have the information that can be used to contribute to control. the review also highlights the evidence for risk of expansion and potential for widespread epidemic and introduction in other regions in Africa in future due to climate change. Furthermore, the review discusses evidence of other species. Nigeria experiences large outbreaks of Lassa fever in the beginning of the year and it has a large impact on preparedness in other African countries. In the article, it describes other species that can be play a role in transmission but invasive rattus species appeared to lower the risk which can be further researched as biological control although different diseases are transmitted by various rodent species. I think the article's content is very important in the current context with the US and foreign international aid because it motivates the control of disease outbreaks in Africa to prevent the exportation of Lassa fever via international travelers as Lassa fever virus can be transmitted between people.
I only find two minor typos in the results
Results
Line 219-220
…in NLigeria…
…to be minimal in December…
This is a well-done review of which the content will direct further research to control Lassa fever better and prevent further spread.
Author Response
Please see attached document with our responses.

Reviewer 2 Report
Comments and Suggestions for Authors
The manuscript provides a comprehensive and well-structured systematic review of environmental drivers influencing the transmission and spread of Lassa virus (LASV) in West Africa. It synthesizes data from 70 studies spanning 1969–2023, highlighting key factors such as seasonality, land-use changes, biodiversity loss, and host range expansion.
The study contributes significantly to understanding the environmental determinants of LASV spread, which is crucial for public health strategies, especially in the face of climate change and land-use alterations. The PRISMA-based methodology, inclusion of multiple databases, and application of predefined eligibility criteria ensure a comprehensive literature selection process. The manuscript effectively highlights the lack of studies on biodiversity loss, the role of secondary hosts, and the influence of climate change on Lassa fever epidemiology. The paper is highly relevant in the context of One Health approaches to disease mitigation, and it effectively addresses knowledge gaps, calling for further interdisciplinary research. However, there are areas where clarity, methodological rigor, and contextual depth could be improved.
For detailed feedback, please refer to the attached file.

Author Response
Please see the attached document with our responses.
